# The VEGF Inhibitor Soluble Fms-like Tyrosine Kinase 1 Does Not Promote AKI-to-CKD Transition

**DOI:** 10.3390/ijms23179660

**Published:** 2022-08-26

**Authors:** Cleo C. L. van Aanhold, Angela Koudijs, Kyra L. Dijkstra, Ron Wolterbeek, Jan A. Bruijn, Cees van Kooten, Hans J. Baelde

**Affiliations:** 1Department of Pathology, Leiden University Medical Center, 2333 ZA Leiden, The Netherlands; 2The Einthoven Laboratory of Vascular and Regenerative Medicine, Division of Nephrology, Department of Internal Medicine, Leiden University Medical Center, 2333 ZA Leiden, The Netherlands; 3Department of Biomedical Data Sciences, Leiden University Medical Center, 2333 ZA Leiden, The Netherlands

**Keywords:** vascular endothelial growth factor, endothelial dysfunction, inflammation, kidney fibrosis

## Abstract

(1) Background: Soluble Fms-like tyrosine kinase 1 (sFLT1) is an endogenous VEGF inhibitor. sFLT1 has been described as an anti-inflammatory treatment for diabetic nephropathy and heart fibrosis. However, sFLT1 has also been related to peritubular capillary (PTC) loss, which promotes fibrogenesis. Here, we studied whether transfection with *sFlt1* aggravates experimental AKI-to-CKD transition and whether sFLT1 is increased in human kidney fibrosis. (2) Methods: Mice were transfected via electroporation with *sFlt1*. After confirming transfection efficacy, mice underwent unilateral ischemia/reperfusion injury (IRI) and were sacrificed 28 days later. Kidney histology and RNA were analyzed to study renal fibrosis, PTC damage and inflammation. Renal sFLT1 mRNA expression was measured in CKD biopsies and control kidney tissue. (3) Results: *sFlt1* transfection did not aggravate renal fibrosis, PTC loss or macrophage recruitment in IRI mice. In contrast, higher transfection efficiency was correlated with reduced expression of pro-fibrotic and pro-inflammatory markers. In the human samples, sFLT1 mRNA levels were similar in CKD and control kidneys and were not correlated with interstitial fibrosis or PTC loss. (4) Conclusion: As we previously found that sFLT1 has therapeutic potential in diabetic nephropathy, our findings indicate that sFLT1 can be administered at a dose that is therapeutically effective in reducing inflammation, without promoting maladaptive kidney damage.

## 1. Introduction

Soluble Fms-like tyrosine kinase 1 (sFLT1) is a circulating splice variant of the vascular endothelial growth factor receptor 1 (VEGFR1/FLT1). sFLT1 spatially controls pro-angiogenic signaling by acting as a VEGF decoy receptor and by forming non-signaling dimers with endothelial VEGF receptors [1], functions that are crucial in vessel sprouting and in maintaining corneal avascularity [2,3]. In addition to these well-established anti-angiogenic effects, sFLT1 also conveys cytoprotective and anti-inflammatory functions. For example, physiological levels of sFLT1 are important for regulating the podocyte actin cytoskeleton [4]. Furthermore, *sFlt1*-deficient mice with heart fibrosis exhibit aggravated tissue fibrosis and an increased recruitment of pro-inflammatory macrophages [5]. Similarly, treatment with low doses of sFLT1 reduces inflammatory disease in preclinical models of diabetic kidney disease, atherosclerosis, heart fibrosis, sepsis, arthritis and atopic dermatitis [5,6,7,8,9,10]. Thus, sFLT1 is a promising anti-inflammatory treatment for several inflammatory conditions.

Apart from these beneficial effects of sFLT1, elevated circulating levels of sFLT1 are also associated with glomerular endothelial injury. This was first studied in preeclampsia, where strongly increased sFLT1 levels are associated with proteinuria, hypertension and renal thrombotic microangiopathy [11]. Since proteinuria and hypertension are recapitulated in patients developing toxicity following treatment with VEGF-inhibiting agents (e.g., anti-VEGF antibodies and Fc-VEGFR-dimers) and in a VEGF deletion model [12,13], the glomerulotoxic effects of excess sFLT1 levels are well supported. To a lower extent, circulating sFLT1 levels are also increased in other renal diseases [14]. For example, during the first week after kidney transplantation and in patients with chronic kidney disease (CKD), serum sFLT1 levels are elevated and are associated with a declined renal function [15,16,17]. As a possible underlying mechanism, higher serum sFLT1 levels are correlated with a lower peritubular capillary number [16]. Although these observations may have been confounded by the kidney condition (e.g., donor age, donor cause of death and ischemia time), these have contributed to a prevailing hypothesis that sFLT1 contributes to renal microvascular injury following acute kidney injury (AKI), thereby promoting the transition towards CKD.

It is currently unknown whether sFLT1 can safely be used therapeutically without aggravating renal microvascular damage. Here, using the same *sFlt1* gene therapy that reverses diabetic kidney disease and dermatitis [6,10], we set out to study the effects of sFLT1 in the unilateral ischemia/reperfusion injury (IRI) model. The kidney IRI model has a well-defined microvascular injury component that is caused by the acute ischemic injury, similar to many forms of human kidney disease observed in patients. In addition, we measured sFLT1 expression in kidneys from patients with CKD and correlated sFLT1 levels with chronic endothelial injury and kidney damage.

## 2. Results

### 2.1. Gene Delivery of sFlt1-VSV and Luciferase

One week before inducing IRI injury, mice were co-transfected with *sFlt1-VSV* and luciferase DNA constructs by bilateral injection of the calf muscles followed by electroporation. The transfection efficiency was monitored weekly by measuring luciferase expression. Luciferase was markedly expressed in transfected mice but not in non-transfected mice (Figure 1A, *p* < 0.0001). The time effect on luciferase expression was not significant (*p* = 0.298), meaning that gene delivery was stable throughout the experiment (Figure 1A). In accordance, IHC staining of muscle tissues obtained 4 weeks after electroporation showed the presence of luciferase-, FLT1- and VSV-positive areas in transfected mice, which were not observed in non-transfected control mice (Figure 1B), confirming robust expression of exogenous VSV-tagged sFLT1 protein. 

### 2.2. sFLT1 Does Not Aggravate Acute and Chronic Kidney Damage following IRI

To determine the effect of sFLT1 on acute and chronic kidney damage, *sFlt1*-transfected and non-transfected mice were subjected to 35 min of unilateral IRI. Serum KIM-1, a marker of proximal tubule injury, was increased to the same degree in sFLT1 and control mice 48 h after IRI (Figure 2B), indicating that the initial proximal tubule injury due to IRI was functionally equivalent in both groups. Furthermore, 28 days after surgery, the IRI kidneys of sFLT1-treated and control mice showed a similar degree of atrophy (Figure 2C). To determine whether sFLT1 treatment affects kidney fibrosis, we analyzed kidney histology and RNA 28 days after IRI. Sirius red staining (Figure 2A,D) and the expression of the fibrotic markers *Col1a1* and *Fn1* (encoding collagen type Iα1 and fibronectin) were markedly increased in IRI mice compared to sham-operated mice (Figure 2A,E,F). sFLT1-treated mice had lower levels of Sirius red-positive staining compared to levels observed in control IRI mice (Figure 2A,D), however, the reduction was not statistically significant (*p* = 0.065). Furthermore, there were no differences in *Col1a1* and *Fn1* mRNA levels between sFLT1-treated and control IRI mice (Figure 2A,E,F). Thus, consistent with our previous findings that sFLT1 has no nephrotoxic effects in type 1 diabetic and *APOC1*-tg mice [6,10], we found that transfection with *sFlt1* does not promote renal fibrogenesis.

### 2.3. sFLT1 Has No Effect on Peritubular Capillary Loss after IRI

To determine whether sFLT1 treatment affects peritubular capillary loss following AKI, we analyzed IHC staining of the IRI kidneys for endomucin on day 28. Endomucin-positive areas in cortex and medulla were reduced in mice that underwent IRI compared to sham mice (Figure 3A,B). However, transfection with *sFLT1* did not affect the endomucin-positive areas compared to non-treated IRI mice (Figure 3A,B). Next, we studied mRNA expression levels of sFLT1’s ligands, the angiogenic factors VEGF (encoded by *Vegfa*) and placental growth factor (encoded by *Pgf*). Renal *Vegfa* mRNA levels were similar between IRI and sham mice (Figure 3C). In contrast, renal *Pgf* levels were significantly higher in the IRI mice compared to the sham mice (Figure 3D). sFLT1 had no significant effect on either renal *Vegfa* or *Pgf* mRNA levels compared to non-treated IRI mice (Figure 3C,D).

### 2.4. sFLT1 Treatment Does Not Alter Macrophage Infiltration after IRI

Compared to sham, IRI surgery increased the recruitment of F4/80-positive macrophages on day 28 (Figure 4A,B). *sFlt1*-transfected mice showed a trend towards an increased F4/80-positive area in the outer medulla compared to control IRI mice, however, this was not significant (Figure 4A,B; *p* = 0.554). Furthermore, compared to sham, the anti-inflammatory macrophage marker CD206 (mannose receptor 1) was increased after IRI (Figure 4A,C). Mice treated with sFLT1 showed a non-significant increase in renal CD206+ macrophages (*p* = 0.122). Consistent with the protein levels, injury-stimulated expression of *Adgre1* and *Mrc1* (encoding F4/80 and CD206) transcripts was similar in sFLT1-treated and control IRI mice (Figure 4D,E). Lastly, renal expression for the monocyte adhesion molecule VCAM-1 (encoded by *Vcam1*) was increased to the same level in sFLT1- and control-treated IRI mice compared to sham (Figure 3A,F).

### 2.5. Transfection Efficiency Correlates with Reduced Kidney Damage and Inflammation

Due to the observed variability in transfection efficiency among *sFlt1*-transfected animals, we next examined whether the transfection efficiency was correlated with renal damage and inflammation. Luciferase expression was not correlated with serum KIM-1 levels measured at day 2 (Table 1), indicating that sFLT1 had no effect on acute tubular injury after IRI. By contrast, consistent with the reduction in renal fibrosis in *sFlt1*-transfected mice (Figure 2C, *p* = 0.065), we found that luciferase expression measured 4 weeks after electroporation negatively correlated with whole-kidney mRNA expression of *Col1a1*, *Col4a2* and *Fn1* (Table 1), meaning that increased sFLT1 delivery correlates with reduced transcription of pro-fibrotic genes. In line with the reduction in collagen transcripts, luciferase expression was also correlated with an attenuation of outer medullary fibrosis (Table 1), however, this was not significant. Notably, transfection efficiency was negatively correlated with mRNA levels of *Pgf* and *Vcam1* (Table 1), factors that mediate the renal recruitment and adhesion of monocytes. As a consequence, transfection efficiency was also negatively correlated with macrophage markers *Adgre1* and *Mrc1* (Table 1). These findings suggest that sFLT1 treatment reduces immune-mediated fibrogenesis following IRI.

### 2.6. Renal sFLT1 mRNA Expression Is Not Associated with Human CKD

The mouse experiment revealed not only that sFLT1 does not aggravate kidney fibrosis and microvascular rarefaction, but also that higher sFLT1 transfection efficiency correlates with reduced immune-mediated fibrogenesis. To test the translational value of these findings, renal sFLT1 transcript levels were measured in biopsies obtained from patients with diabetes-related CKD and control kidney tissues [18]. 

Gene expression analysis revealed that renal *sFLT1-i13* (encodes sFLT1 isoform 2) and *sFLT1-e15a* (encodes sFLT1 isoform 3) levels measured in CKD kidneys were similar to levels found in control cases (Figure 5A,B). Renal *sFLT1-e15a* mRNA expression was generally very low, making it unlikely to play a role mechanistically. We next examined whether sFLT1-i13 transcript levels correlate with histological parameters and expressed genes previously documented in this cohort [18]. In CKD kidneys, sFLT1-i13 levels were not correlated with interstitial fibrosis (Figure 5C) or the interstitial CD31-positive area (Figure 5D). However, sFLT1-i13 levels were positively correlated with thrombomodulin mRNA levels (Figure 5E), an endothelial anti-inflammatory protein [19]. In addition, sFLT1-13 transcript levels were negatively correlated with syndecan-1 mRNA levels (Figure 5F), a mediator of renal fibrogenesis [20].

## 3. Discussion

Here, we show that transfection with the gene encoding the VEGF inhibitor sFLT1 does not promote AKI-to-CKD transition. Transfection with *sFlt1* did not aggravate renal fibrosis, decrease the peritubular capillary number or increase the accumulation of renal macrophages following IRI. In contrast, higher transfection efficiency levels were associated with a reduced renal expression of pro-fibrotic and pro-inflammatory factors. Moreover, in patients with CKD, renal sFLT1 levels were not increased compared to healthy control kidneys, and renal sFLT1 levels measured in CKD biopsies were not correlated with a loss of renal endothelium or renal damage. Prior studies found that transfection with *sFlt1* has therapeutic potential in diabetic kidney disease and several inflammatory conditions. Thus, our findings indicate that sFLT1 can be administered using a method that is therapeutically effective in reducing inflammation, without aggravating the progression of kidney damage.

We have reproduced the finding that *sFlt1* transfection has no deleterious effects on renal endothelial function in several independent models of kidney fibrosis; in the IRI model presented here, in a streptozotocin-induced DKD model [6] and in *APOC1*-tg mice [10] that develop glomerulosclerosis [21], transfection with *sFlt1* did not drive renal damage or dysfunction. Contrary to our results, treatment with VEGF inhibitors has widely been associated with renal side effects. For example, a deficiency in glomerular VEGF results in development of proteinuria, hypertension, severe glomerular endothelial cell injury and thrombotic microangiopathy [13]. Similarly, treating mice with different anti-VEGF compounds causes proteinuria and renal dysfunction [22]. These opposite observations suggest that our *sFlt1* transfection is quite different from anti-VEGF strategies used in other studies. First, the relatively low systemic sFLT1 levels achieved by transfection in our studies may have been insufficient to ablate VEGF signaling and to induce endothelial injury. Prior studies that used low-dose administration of FLT1 domain 1–3 (e.g., 400 ng daily) demonstrated profound anti-inflammatory effects without inducing renal side effects [5,7], while higher doses of a sFLT1-Fc chimera protein (300 ng/h) induced endothelial rarefaction and kidney fibrosis [16]. In line with this, Sugimoto et al. [22] found that administration with an sFLT1-Fc chimera protein causes proteinuria only when the achieved sFLT1 levels exceed physiological VEGF levels. Together, these studies indicate that markedly increased sFLT1 levels are needed to induce glomerular endothelial injury. Notably, *sFlt1* transfection did not aggravate renal damage or endothelial injury, thus underscoring that VEGF levels remained within the physiological range, thereby allowing maintenance of endothelial cell function. Second, different types of VEGF inhibition (e.g., native sFLT1, sFLT1-dimer, anti-VEGF mAb, sVEGFR2, VEGF-trap) may achieve different extents of VEGF ablation and renal side effects. For example, “VEGF-trap”, a FLT1/KDR-Fc fusion protein, more efficiently sequesters VEGF than the native full-length sFLT1 [23], which we used in our studies. In addition, native sFLT1, but not fusion or multimer FLT1 variants, participates in additional homotypic interactions with VEGF receptors [24], of which the effects are still to be studied. Therefore, it is likely that sFLT1 modulates VEGF signaling using mechanisms beyond sequestering VEGF, possibly indicating that an sFLT1-based therapy is less nephrotoxic than therapies based on anti-VEGF antibodies or dimerized proteins, which only sequester VEGF. Thus, the role of sFLT1 in the regression or progression of kidney injury is highly dose- and compound-dependent, exerting opposing roles, from renoprotection by reducing inflammation to nephrotoxicity by perturbing glomerular endothelial function. Future investigation is needed to define the delicate therapeutic window of sFLT1, and to characterize the cellular and molecular interactions that on the one hand underlie sFLT1’s anti-inflammatory properties, and at high levels underlie sFLT1’s renal side effects.

We did not only find that sFLT1 does not aggravate the development of kidney injury in the IRI model; in contrast, sFLT1-treated mice seemed to develop less renal fibrosis compared to control mice that underwent IRI (Figure 2C). On top of that, treatment efficiency was inversely correlated with levels of pro-fibrotic and pro-inflammatory factors. The findings suggest that sFLT1 treatment modestly reduces immune-mediated fibrosis progression following IRI. In support of the potential benefits of VEGF inhibition following IRI, Lin et al. [25] previously showed that transfection with sVEGFR2 ameliorated peritubular capillary loss, inflammation, pericyte differentiation and fibrosis in both a unilateral IRI model and in a ureteral obstruction model. As a mechanism of action, the authors found that an increased production of VEGF by pericytes, myofibroblasts and inflammatory macrophages after ureteral obstruction promotes renal inflammation and fibrosis [25]. In support of this, previous studies found that sFLT1 reduces inflammatory conditions by reducing VEGF-induced endothelial hyperpermeability and adhesion molecule expression, reducing the chemoattraction of FLT1-expressing monocytes and macrophages and by decreasing PGF-dependent MCP-1 expression [5,6,7,8,9,10]. Therefore, sFLT1 may reduce immune-mediated fibrogenesis by inhibiting VEGF- and PGF-dependent endothelial cell activation and inflammation. The absence of a larger treatment effect may have several explanations. First, there was technical variability in both the IRI model and in the transfection technique, which are addressed by measurement of KIM-1 levels and luciferase expression. Improved reproducibility of the IRI model and/or of the transfection technique might result in a significant reduction in renal fibrosis. Second, although there was a considerable increase in renal *Pgf* mRNA levels following IRI, we did not find an increase in *Vegfa* mRNA levels in IRI kidneys compared to sham. This may imply that, while the correlations between higher sFLT1 levels and reduced fibrosis could be explained by inhibited PGF, the lack of a larger treatment effect could also be explained by a lack of inhibitable VEGF.

Regarding the measurement of sFLT1 in human CKD biopsies, we found no difference in mRNA expression of sFLT1-i13 and sFLT1-e15a between CKD and control kidneys; in addition, renal sFlt1 transcript levels were not correlated with interstitial fibrosis or peritubular capillary rarefaction. However, sFLT1 transcript levels were positively correlated with expression of thrombomodulin, an endothelial receptor that protects against renal cell death and inflammation [19]. In addition, sFlt1 levels were negatively correlated with syndecan-1 levels, a protein known to mediate renal fibrogenesis [20], although its exact role in CKD is not completely understood yet. In line with our mouse experiment, these observations suggest that renal sFLT1 is produced locally as a compensatory mechanism to reduce inflammation and fibrosis. Consistent with this, Onoue et al. [7] identified that renal sFlt1-i13 mRNA levels were positively correlated with renal function in CKD patients. Paradoxically, Di Marco et al. [17] found that serum sFLT1 levels were increased in CKD patients and were negatively correlated with renal function. However, when serum sFLT1 levels were measured following heparin administration—also taking into account heparan sulfate-bound sFLT1 that is stored in the glycocalyx—total sFLT1 levels were lower in CKD patients compared to healthy controls, and were positively correlated with renal function [26]. Together with our observation that renal sFLT1 synthesis is unaffected in CKD, these studies suggest that changes in the endothelial glycocalyx [27] render CKD patients unable to store sFLT1 and to protect it from clearance, thereby decreasing the total sFLT1 reservoir. Further investigation using conditional expression models will be required to determine whether the loss of bioavailable sFLT1 in CKD is causally related to kidney disease progression.

In conclusion, sFLT1 is a viable treatment for chronic inflammatory conditions such as diabetic kidney disease, heart fibrosis and eczema. Although sFLT1 is typically associated with renal side effects, our results demonstrate that low-dose treatment with native sFLT1 does not drive endothelial injury to further promote progressive fibrosis. The current findings indicate that sFLT1 can safely be used to limit inflammatory conditions without promoting kidney fibrosis.

## 4. Materials and Methods

### 4.1. Animals

All animal experiments were conducted in accordance with the guidelines of the animal welfare committee of the Dutch Animal Experiments Committee (AVD license 11600202010445) and the Leiden University Medical Center. Male 7-week-old C57Bl/6 mice were purchased from Charles River Laboratories (Wilmington, MA, USA); male mice were exclusively used to reduce total numbers of mice required for statistical analysis due to the substantial difference in susceptibility to IRI between male and female mice [28]. Upon arrival, mice were randomized to group-specific cages. Mice were housed under standard laboratory conditions with free access to water and standard rodent chow, and were sacrificed by means of cervical dislocation under anesthesia with isoflurane. Serum samples were collected at sacrifice. One mouse in the transfected IRI group was removed from the study before IRI surgery due to the presence of hydronephrosis.

### 4.2. sFlt1-VSV Transfection

The transfection protocol was described previously with one adjustment (40 μg *sFlt1-VSV* DNA instead of 20 μg per muscle) [6,29]. In brief, two pcDNA3.1 vectors (Invitrogen, Breda, the Netherlands) were constructed, containing either mouse *sFlt1-VSV* or the luciferase gene, driven by the same cytomegalovirus promoter. The plasmids were amplified in Escherichia coli DH5α (Invitrogen), purified using the QIAfilter Plasmid Maxi-prep kit (Qiagen, Venlo, The Netherlands) and dissolved in EndoFree Tris-EDTA buffer (Qiagen). The mice were co-transfected by electroporation with the *sFlt1-VSV* and luciferase constructs (40:1) in both gastrocnemius muscles (40 μg DNA each). To monitor transfection efficiency, the mice were anesthetized and injected intraperitoneally with D-luciferin (150 mg/kg; Synchem OHG, Altenburg, Germany) weekly. Five minutes after the luciferin injection, luciferase activity was visualized using a NightOWL bioluminescence camera (Xenogen Ivis Spectrum, Alameda, CA, USA).

### 4.3. Unilateral IRI Model

Renal unilateral IRI was induced 7 days after transfection (after confirming transfection efficiency). Mice were subjected to anesthesia by inhalation of isoflurane (induction: 4%, 0.7 L/min; maintenance: 2%, 0.5 L/min) and prolonged analgesia by 3 subcutaneous injections with buprenorphine (0.1 mg/kg, before and after surgery, morning after surgery) supplemented with buprenorphine in the drinking water (7 μg/mL) for 48 h, starting 24 h before surgery. Unilateral IRI was achieved via a dorsal back incision allowing clamping of the left renal pedicle using a non-traumatic microaneurysm clip for 35 min on a 37 °C probe-controlled homeothermic heating pad, leaving the right kidney intact. Sham operation was performed by dorsal back incision only. During surgery, the mice were intraperitoneally injected with PBS to avoid dehydration. Mice were killed at day 28 after surgery by exsanguination via heart puncture under anesthesia and analgesia. Blood, kidney and muscle tissue samples were obtained for further analysis.

### 4.4. ELISA of Serum Kidney Injury Molecule-1

Mouse blood was drawn 48 h after IRI surgery. Serum kidney injury molecule-1 (KIM-1) concentrations were measured using the mouse TIM-1/KIM-1/HAVCR Quantikine ELISA Kit (R&D Systems, Minneapolis, MN, USA) according to the manufacturer’s instructions.

### 4.5. Histology and Immunohistochemistry

Kidneys were fixed in 10% formalin and embedded in paraffin. For detection of Sirius red-positive collagen, deparaffinized kidney sections (4 μm) were rehydrated, predifferentiated with 0.2% phosphomolybdic acid for 5 min, stained with 0.1% picrosirius red (Direct Red 80, REF: 365548; Sigma-Aldrich, St. Louis, MO, USA) in picric acid for 90 min and differentiated with saturated picric acid. Primary antibodies against fibronectin (1:4800, F3648, Sigma), collagen type IV (1:500, ab19808, Abcam, Cambridge, UK), endomucin (1:20,000, 14-5851-82, EBioScience, San Diego, CA), VCAM-1 (1:800, 553330, BD Biosciences, Breda, The Netherlands), F4/80 (1:100; kindly provided by the Department of Human Genetics, Leiden University Medical Center, Leiden, The Netherlands) and CD206 (1:2000, ab64693, Abcam) were used for immunohistochemistry (IHC) of paraffin-embedded tissues. Frozen sections were fixed with acetone/ethanol and immunostained with primary antibodies against VSV-G (1:2000, V4888, Sigma-Aldrich), luciferase (1:3000, ab181640, Abcam), FLT-1 (1:9000, AF471, R&D Systems) and VCAM-1 (1:800, 553330, BD Biosciences). The appropriate Envision (Dako), Impress (Vector Laboratories) or Goat-on-Rodent (BioCare) HRP-conjugated secondary reagents were used with DAB+ as the chromogen. Non-specific isotype-matched antibodies were used as a negative control.

### 4.6. Digital Image Analysis

Tissue slides were scanned using an IMS slide scanner (Philips). Six independent fields in the cortex and four independent fields in the outer medulla were analyzed per kidney at 40× magnification, and the percent area of Sirius red-positive staining and endomucin-, F4/80- or CD206-positive immunostaining was quantified using ImageJ (NIH, Bethesda, MD, USA). For measuring the Sirius red-positive area, glomeruli and large blood vessels were digitally removed from the high-power fields.

### 4.7. Quantitative PCR

To quantify changes in gene expression, total RNA was extracted from 10 frozen kidney sections (10 µm) using TRIzol extraction buffer (ThermoFisher Scientific, Waltham, MA, USA) and converted to cDNA with AMV reverse transcriptase (Roche) using random hexamer primers. Quantitative real-time PCR was performed using IQ SYBR Green Supermix (Bio-Rad, Hercules, CA, USA) on a Bio-Rad CFX real-time system. Cycle threshold values were normalized to the housekeeping gene *Hprt1*. The primer sets in Table 2 were used in this study.

### 4.8. Human CKD Biopsy Cohort

A biopsy cohort, previously documented by Baelde et al. [18], was used to correlate whole-cortical mRNA expression with histological data. The clinical and histopathological characteristics are summarized in Table 3. In short, fresh-frozen biopsies were obtained from patients with diabetes-related CKD (*n* = 30). CKD was histologically confirmed in periodic acid-Schiff stained sections. As a control group (*n* = 10), deceased donor kidneys unsuitable for transplantation for technical reasons (*n* = 6), native kidneys with normal function obtained at autopsy (*n* = 3) and the non-affected part of a tumor-nephrectomy sample (*n* = 1) were included. There were no significant differences between the different control subgroups for any of the clinical parameters. Furthermore, in deceased donor kidneys, there was no presence of acute tubular necrosis. Cases were handled in accordance with Dutch national ethics guidelines and in accordance with the Code of Conduct regarding the Proper Secondary Use of Human Tissue.

### 4.9. Statistical Analysis

Data are expressed as means ± SD. Multigroup comparison was performed using one-way ANOVA followed by the LSD multiple comparison test for subgroup comparison if the overall F-test was significant, or, in the event of unequal variances, followed by appropriate adjustments according to Tamhane’s procedure; to analyze differences in transfection efficiency and positively stained areas between groups, linear mixed models that take into account the correlation of observations within mice were used; and the Pearson correlation coefficient r was determined for correlations using SPSS. Luciferase expression, positively stained areas and normalized gene expression data were log-transformed or arcsine-transformed, depending on the shape of the histogram of the residuals. An alpha level of 0.05 was used to assess statistical significance.

## Figures and Tables

**Figure 1 ijms-23-09660-f001:**
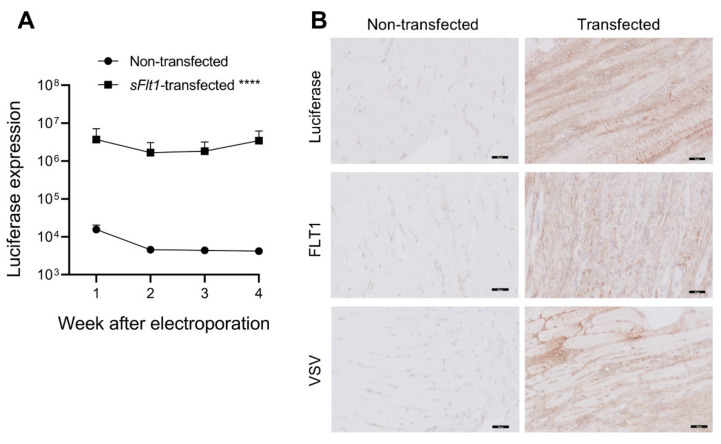
Transfection efficiency. (**A**) The *sFlt1-VSV* and luciferase constructs were co-transfected by injection into the calf muscles. Following an intraperitoneal injection of luciferin, luciferase expression was measured using a NightOWL bioluminescence camera. Measurement of the total flux (photons/s) showed efficient transfection at all indicated timepoints. **** *p* < 0.0001 versus non-transfected mice, linear mixed model using log-transformed luciferase levels. (**B**) Representative images of luciferase, FLT1 and VSV-G immunostaining of calf muscle tissues 4 weeks after electroporation shows positively stained areas in transfected mice, but not in non-transfected mice.

**Figure 2 ijms-23-09660-f002:**
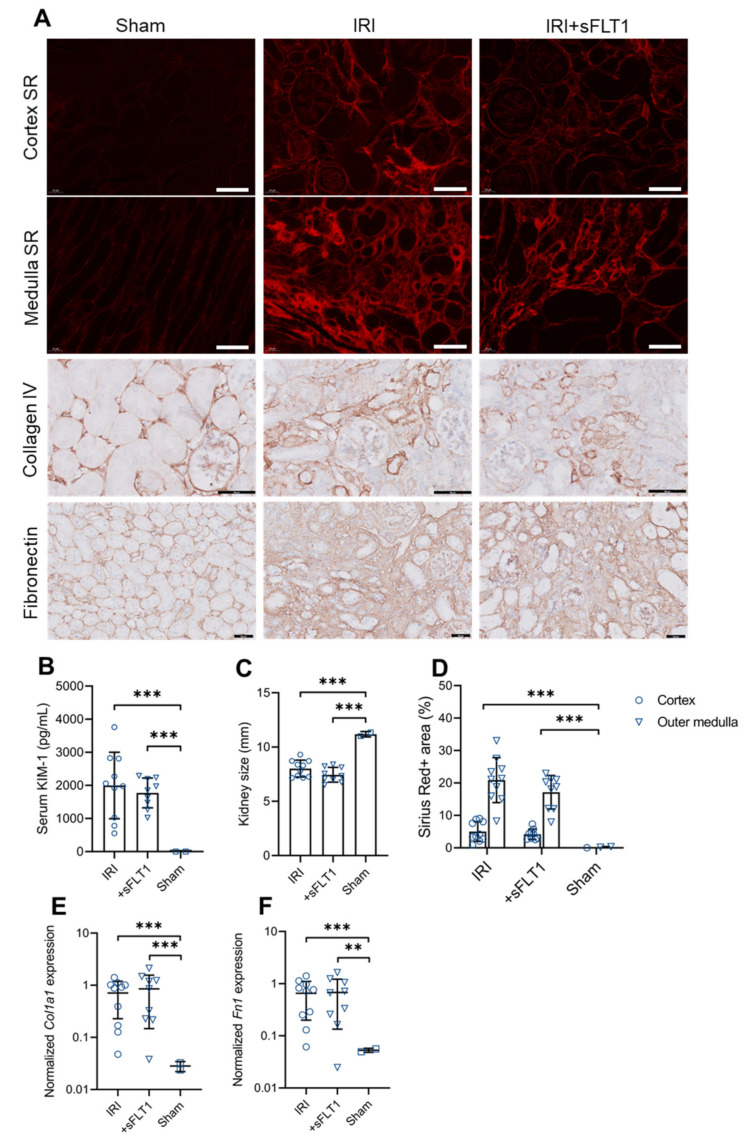
sFLT1 does not accelerate AKI-to-CKD progression. One week after transfection, control and *sFlt1*-transfected mice were subjected to 35 min of warm unilateral IRI. IRI and sham kidneys were harvested 28 days after surgery. (**A**) Representative images of Sirius red (SR) in renal cortex and outer medulla, collagen IV and fibronectin 28 days after IRI. Scale bars, 50 µm. (**B**) Serum KIM-1 levels of mice were measured 48 h after IRI. One-way ANOVA. (**C**) the kidney pole-to-pole size was measured 28 days after IRI. One-way ANOVA. (**D**) The Sirius red-positive cortex and outer medulla areas were quantified. Linear mixed model analysis. (**E**,**F**) Quantitative PCR for (**E**) *Col1a1* and (**F**) *Fn1* was performed on whole-kidney mRNA. One-way ANOVA on log-transformed gene expression. **** *p* < 0.001, ** *p* < 0.01.

**Figure 3 ijms-23-09660-f003:**
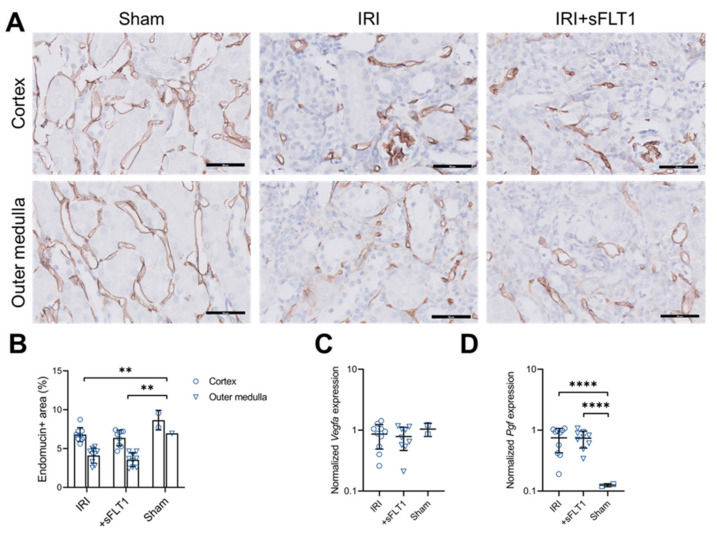
sFLT1 has no effect on chronic endothelial injury after IRI. (**A**,**B**) Kidney sections were immunostained with anti-endomucin as shown in (**A**) and positively stained areas were quantified in cortex and outer medulla (**B**). Scale bars, 50 µm. Linear mixed model. Quantitative PCR for (**C**) *Vegfa* and (**D**) *Pgf* was performed on whole-kidney mRNA. One-way ANOVA, *** *p* < 0.001, ** *p* < 0.01.

**Figure 4 ijms-23-09660-f004:**
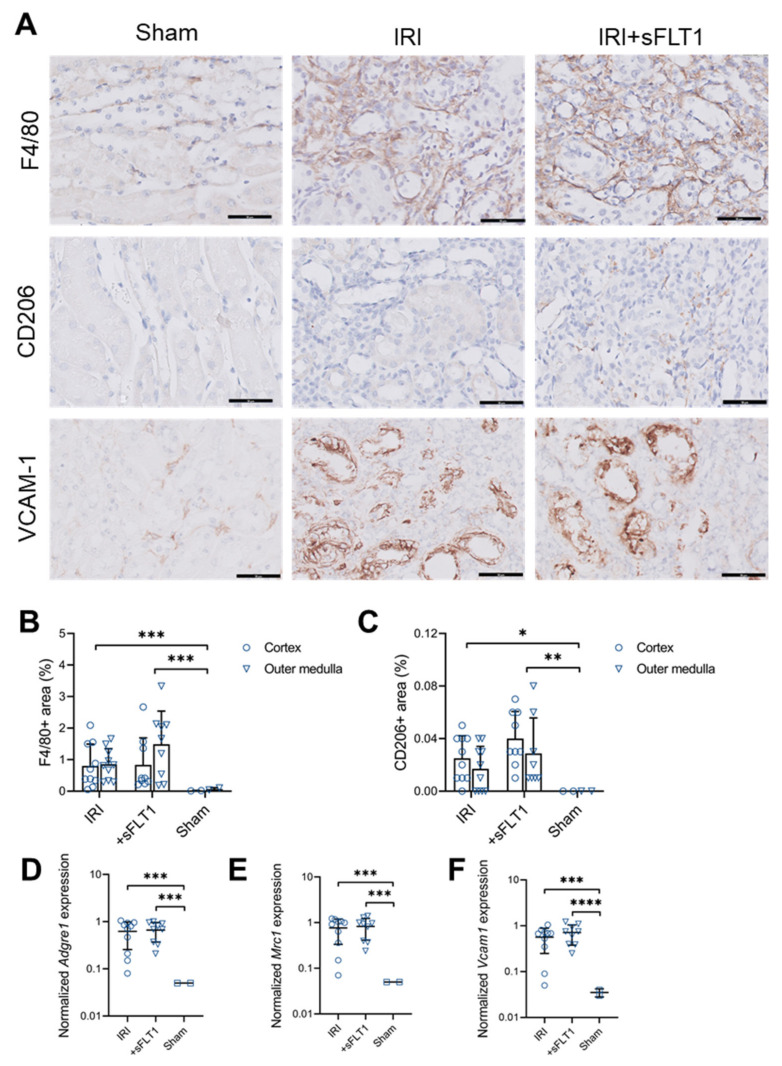
sFLT1 does not affect chronic macrophage infiltration after IRI. (**A**) Representative images of renal F4/80, CD206 and VCAM-1. Scale bars, 50 µm. (**B**) F4/80-positive areas were quantified in cortex and outer medulla. Linear mixed model using log-transformed values. (**C**) CD206-positive areas were quantified in cortex and outer medulla. Linear mixed model using arcsine-transformed CD206 data. Quantitative PCR for (**D**) *Adgre1*, (**E**) *Mrc1* and (**F**) *Vcam1* was performed on whole-kidney mRNA. One-way ANOVA, **** *p* < 0.0001, *** *p* < 0.001, ** *p* < 0.01, * *p* < 0.05.

**Figure 5 ijms-23-09660-f005:**
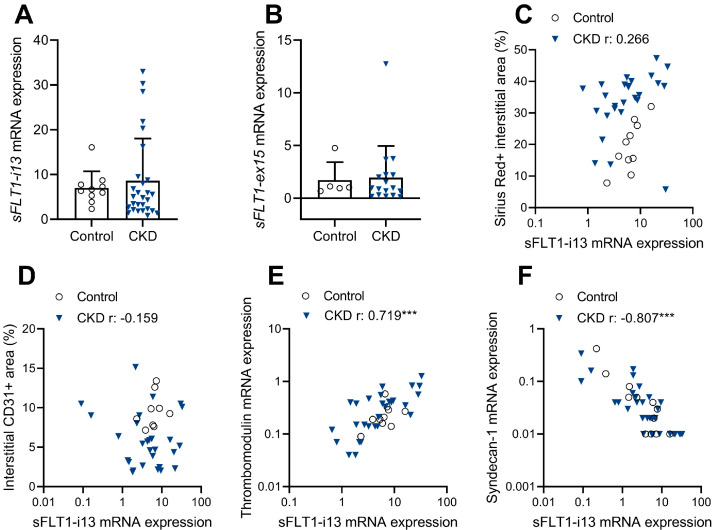
Renal sFLT1 transcript levels are not associated with human CKD. Renal *sFLT1-i13* and *sFLT1-e15a* transcript levels were measured in fibrotic kidney biopsies obtained from patients with diabetes-related CKD (CKD) and in control kidney tissues. (**A**) Renal *sFLT1-i13* and (**B**) *sFLT1-e15a* mRNA levels are not increased in CKD kidneys compared to control kidneys. Independent samples *t*-test of log-transformed mRNA expression. Correlation plots of renal *sFLT1-i13* gene expression with (**C**) Sirius red staining, (**D**) CD31 immunostaining, (**E**) thrombomodulin mRNA expression and (**F**) syndecan-1 mRNA expression split per group. Bivariate Pearson correlations of log-transformed mRNA levels. *** *p* < 0.001.

**Table 1 ijms-23-09660-t001:** Bivariate correlations between transfection efficiency and parameters in sFLT1-treated mice.

	Bivariate Correlation with Luciferase Expression	
	Pearson’s r	*p*-Value
Serum KIM-1 day 2	0.626	0.071
Kidney size (mm)	0.394	0.295
Outer medulla Sirius red	−0.514	0.157
Outer medulla endomucin	0.053	0.893
Outer medulla F4/80	−0.091	0.817
*Col1a1*	−0.751	0.020
*Col4a2*	−0.756	0.018
*Fn1*	−0.759	0.018
*Pgf*	−0.889	0.001
*Adgre1*	−0.863	0.003
*Mrc1*	−0.884	0.002
*Il1b*	0.074	0.850
*Vegfa*	−0.667	0.050
*Vcam1*	−0.839	0.005

The log-transformed luciferase expression measured at week 4 after electroporation was correlated with the indicated parameters (*n* = 9 animals). Normalized mRNA expression values were log-transformed.

**Table 2 ijms-23-09660-t002:** Primer sequences for real-time PCR analysis.

Target	Forward	Reverse
*Hprt*	AGATGGTCAAGGTCGCAAGC	TCAAGGGCATATCCTACAACAAAC
*Col1a1*	TGACTGGAAGAGCGGAGAGT	AGACGGCTGAGTAGGGAACA
*Fn1*	GGCAGGCTCAGCAAATCG	CATAGCAGGTACAAACCAGGG
*Col4a1*	CTGGGATCATGGACCGAGTG	CCTTTCTCCGGGTAGCACTG
*Vegfa*	CTGGACCCTGGCTTTACTGC’	GCTTCGCTGGTAGACATCCA
*Pgf*	TTCTGGAGACGACAAAGGCA	GCTGGTTACCTCCGGGAAAT
*Adgre1*	GGCAGGGATCTTGGTTATGCT	GCTGCACTCTGTAAGGACACT
*Mrc1*	GCTGGCGAGCATCAAGAGTA	CATCACTCCAGGTGAACCCC
*Vcam1*	GAAATGCCACCCTCACCTTA	ACGTCAGAACAACCGAATCC
*GAPDH*	CGACCACTTTGTCAAGCTCA	AGGGGTCTACATGGCAACTG
*sFLT1-i13*	ACAATCAGAGGTGAGCACTGCAA	TCCGAGCCTGAAAGTTAGCAA
*sFLT1-e15a*	CGAGCCTCAGATCACTTGGT	CGATGACGATGGTGACGTT
*THBD*	ACATCCTGGACGACGGTTTC	CGCAGATGCACTCGAAGGTA
*SDC1*	TGCCGCAAATTGTGGCTACTAAT	GAGCTGCGTGTCCTTCCAAG

The log-transformed luciferase expression measured at week 4 after electroporation was correlated with the indicated parameters (*n* = 9 animals). Normalized mRNA expression values were log-transformed.

**Table 3 ijms-23-09660-t003:** Clinical and histopathological characteristics of controls and diabetic CKD cases.

Clinical Characteristics	Controls (*n* = 10)	CKD (*n* = 30)	*p* Value
Age (years)	58.3 ± 14.2	59.9 ± 9.8	0.698
Female, *n* (%)	3 (30)	15 (50)	0.271
Hypertension, *n* (%)	3/7 (43)	14/20 (70)	0.201
Systolic blood pressure, mmHg	135 ± 31	137 ± 17	0.884
Diastolic blood pressure, mmHg	74 ± 16	80 ± 10	0.329
Serum creatinine, mmol/L	81 ± 48	202 ± 138	0.015
Proteinuria, g/L	0.03 ± 0.05	3.7 ± 1.9	<0.001
**Histopathological characteristics**	**Controls (*n* = 10)**	**CKD (*n* = 28)**	***p* value**
Glomerulosclerosis, *n* (%)			
Absent	10 (100)	0	
Diffuse		11 (39)	
Diffuse and nodular		2 (7)	
Nodular		15 (54)	
FSGS		1 (4)	
Tubular atrophy		13 (46)	
Interstitial fibrosis		12 (43)	
Arteriosclerosis		14 (50)	
Arteriolar hyalinosis		2 (7)	
IgA GN		2 (7)	
Membranous GN		1 (4)	
Interstitial Sirius red (%)	19.5 ± 7.8	33.3 ± 9.8	<0.001
Interstitial CD31 (%)	9.6 ± 2.2	5.5 ± 3.4	0.002

FSGS, focal segmental glomerulosclerosis. GN, glomerulonephritis.

## Data Availability

The data presented in this study are available on request from the corresponding author.

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
