# Peer review of "The VEGF Inhibitor Soluble Fms-like Tyrosine Kinase 1 Does Not Promote AKI-to-CKD Transition"

_ijms, 2022, doi:10.3390/ijms23179660_

Round 1

Reviewer 1 Report

The article entitled “The VEGF inhibitor Soluble Fms-like Tyrosine Kinase 1 does not promote AKI-to-CKD transition” evaluated the effects of sFLT1, a circulating splice variant of the VEGFR1 and a key regulator of the angiogenic signaling pathway. The main findings comprised the absence of fibrosis, PTC loss, and macrophage infiltration after sFLT1 treatment. Importantly, sFLT1 correlated with reduced expression of profibrotic and proinflammatory markers. Likewise, in human samples, sFLT1 levels were similar in CKD and control kidneys and were not correlated with interstitial fibrosis or PTC loss. The study is interesting and contributes to knowledge in the field. Please address the following comments before we proceed.

     Major comments

1)    Please provide more information on human samples, such as age and sex. Were demographics comparable between diabetic individuals, kidney donors, and patients undergoing nephrectomy?

2)    Likewise, why did diabetic individuals undergo renal biopsy? Were the histological findings consistent with diabetic kidney disease? Or have other findings been documented?

3)    Was acute tubular necrosis found despite discarding kidney donors due to technical reasons? Or the histological finding was merely kidney fibrosis? Please clarify.                  

Minor comment

4)    Replace “cadaver donor kidneys” for “deceased donor kidneys” (page 12, line 381), as this term is more appropriate.  

Author Response

Dear reviewer,

Thank you for the helpful comments to our manuscript. Please see the attachment for our point-by-point response. 

Sincerely,

Cleo van Aanhold

Reviewer 2 Report

The paper, The VEGF inhibitor Soluble Fms-like Tyrosine Kinase 1 does 2 not promote AKI-to-CKD transition, is a fairly well executed and written demonstration of the non-fibrogenic effect of sFLT1 in an acute kidney injury model.

As to writing, please spell out abbreviations at first use: eg IRI and AKI; also mention the proteins produced by referenced genes like, ColA1 AND Fn1 (line 100 instead of line 157). 

It looks as if reference 6 is wrong.  I think it is supposed to be Bus et al, J Pathol2017;241:589–599. (line 101 and others).

As to the work, it would be more convincing if IHC was also provided for Collagen 1 (Col1a1) and fibronectin (Fn1) in Figure 2,on the basis that some readers may think that protein levels and mRNA levels do not always correlate.  The same for Vcam1 in Fig. 4.

Author Response

(The authors gave the same response as above.)
